# Seprevalence and correlates of SARS-CoV-2 neutralizing antibodies from a population-based study in Bonn, Germany

N. Ahmad Aziz [1,2], Victor M. Corman [3,4], Antje K. C. Echterhoff[1], Marcel A. Müller [3,4], Anja Richter[3], Antonio Schmandke[1], Marie Luisa Schmidt[3,4], Thomas H. Schmidt [1], Folgerdiena M. de Vries[1], Christian Drosten [3,4] & Monique M. B. Breteler [1,5✉]

To estimate the seroprevalence and temporal course of SARS-CoV-2 neutralizing antibodies, we embedded a multi-tiered seroprevalence survey within an ongoing community-based cohort study in Bonn, Germany. We first assessed anti-SARS-CoV-2 immunoglobulin G levels with an immunoassay, followed by confirmatory testing of borderline and positive test results with a recombinant spike-based immunofluorescence assay and a plaque reduction neutralization test (PRNT). Those with a borderline or positive immunoassay result were retested after 4 to 5 months. At baseline, 4771 persons participated (88% response rate). Between April 24th and June 30th, 2020, seroprevalence was 0.97% (95% CI: 0.72−1.30) by immunoassay and 0.36% (95% CI: 0.21−0.61) when considering only those with two additional positive confirmatory tests. Importantly, about 20% of PRNT+ individuals lost their neutralizing antibodies within five months. Here, we show that neutralizing antibodies are detectable in only one third of those with a positive immunoassay result, and wane relatively quickly.

[1] Population Health Sciences, German Center for Neurodegenerative Diseases (DZNE), Bonn, Germany. [2] Department of Neurology, Faculty of Medicine, University of Bonn, Bonn, Germany. [3] Institute of Virology, Charité-Universitätsmedizin Berlin, corporate member of Freie Universität Berlin, Humboldt-Universität zu Berlin, Berlin Institute of Health, Berlin, Germany. [4] German Center for Infection Research (DZIF), Berlin, Germany. [5] Institute for Medical Biometry, Informatics and Epidemiology (IMBIE), Faculty of Medicine, University of Bonn, Bonn, Germany. ✉email: monique.breteler@dzne.de

As of January 20th 2021, the severe acute respiratory syndrome coronavirus-2 (SARS-CoV-2) has infected more than 96 million people worldwide, resulting in more than two million deaths due to coronavirus disease 2019 (COVID-19)[1,2]. Even after the recent development of highly effective vaccines, accurate estimates of SARS-CoV-2 seroprevalence patterns in the general population remain essential for containing the still raging pandemic as logistic challenges of manufacturing, delivery and deployment, as well as preconceptions against vaccination, are expected to severely delay their timely and widespread administration[3–5]. Moreover, still much remains unknown about the efficacy of currently available vaccines against emerging SARS-CoV-2 strains[6]. Seroprevalence, correlates and temporal dynamics of SARS-CoV-2 neutralizing antibodies are crucial for gauging population immunity, but have hardly been investigated in seroepidemiological studies to date[7–9]. Population studies are the only way to gain knowledge about the prevalence of asymptomatic and mildly symptomatic cases, as well as the potency and sustainability of their acquired immune response, both of which are of paramount importance from a public health perspective, because such individuals often elude the classical symptom-based infection chain tracking, but yet play a key role in the further spreading and sustainment of the current global outbreak[10]. Moreover, seroprevalence studies provide important benchmarks for tracking the evolution of the pandemic by enabling incidence estimates at population-level[11].

Many population-based SARS-CoV-2 serosurveillance studies have already been performed around the globe with widely varying seroprevalence estimates[12–20]. Many of these estimates may have been biased[17], due to inadequate sampling methods, poor antibody test performance, non-random sampling (e.g., self-referral), a non-representative sampling setting (e.g., blood donors and hospital workers), as well as small sample sizes[17]. Especially, studies that compare seroprevalence estimates based on multiple sampling approaches covering the same geographical region and timeframe, while using standardized serological assays, are urgently needed as these could enable a better estimation of the magnitude of bias engendered by using convenience sampling methods, that are currently the main source of seroprevalence estimates for most parts of the world[21]. In Germany, while several seroprevalence studies are still ongoing, findings of only a few serosurveys have been published. Only three of these were population-based studies[22–24], while the remaining assessments targeted industrial workers, health-care providers, mothers, students/teachers, or blood donors[18]. COVID-19-related morbidity and mortality rates have been relatively low in Germany compared to other (European) countries[25,26], yet the true exposure state of the population could be much higher given the unknown proportion of SARS-CoV-2 infections with mild or asymptomatic course[15].

An important challenge for accurate assessment of SARS-CoV-2 seroprevalence, especially in regions with a relatively low infection rate, is insufficient specificity of the serological tests. Widely used (point-of-care) lateral flow and quantitative ELISAs lead to a relatively large number of false positives due to cross-reactivity with other (endemic) coronaviruses[27]. The current gold standard for SARS-CoV-2 serology are neutralization assays[27]. The presence of antibodies that can neutralize the virus is highly specific for having sustained an infection, and is also thought to constitute a major component of the acquired immunity to the virus[28]. However, neutralization assays are highly laborious, can only be performed in specialized (i.e., biosafety level-3) laboratories, and hitherto have hardly been used in serosurveys. Thus, currently little is known about the determinants, correlates and temporal evolution of SARS-CoV-2 neutralizing antibodies at population-level.

We aimed to (1) accurately estimate the prevalence of SARS-CoV-2 seropositivity in a region with a relatively low infection rate, using a multitiered serological testing strategy including highly specific neutralization assays, and (2) investigate the correlates and temporal dynamics of neutralizing antibodies against SARS-CoV-2, with a particular focus on the characteristics of infected but asymptomatic or mildly/moderately symptomatic individuals. By embedding a large-scale seroepidemiological study within the pre-existing framework of an ongoing prospective community-based cohort study, we intended to prevent self-referral bias, ensure long-term follow-up of the participants (including future seroconversion), and facilitate the investigation of genetic, health, and lifestyle determinants of susceptibility and resilience to SARS-CoV-2 infection. In this report, we present the findings of the first serosurvey including follow-up of all individuals with a neutralizing antibody response against SARS-CoV-2.

## Results

**Seroprevalence estimates.** Cohort characteristics and the serosurvey results of Group I are displayed in Table 1 and Fig. 1. The participants originated from a total of 3983 different households, including 778 households with two and five households with three participants. 16 of the 46 individuals with a positive enzyme-linked immunosorbent assays (ELISA) result had neutralizing antibodies, whereas this was the case for only one of the 36 persons with an ELISA result within the borderline range (Fig. 1).

The prevalence estimate based on ELISA results only, an approach that prioritizes sensitivity and thereby provides an estimate of the upper bound of the seroprevalence, was 0.97% (95% confidence interval (CI): 0.72–1.30). When only treating those participants as cases who had tested positive on all three tests (including one individual who had a borderline ELISA test result, but tested positive on both the recombinant immunofluorescence test and the plaque reduction neutralization test (PRNT), see Fig. 1), an approach that prioritizes specificity and

**Table 1 Sample characteristics (Group I) stratified by serostatus.**

| Serostatus | N | Age (y) | Sex (f) | Education (high) | BMI | Number of comorbidities | Number of symptoms |
|---|---|---|---|---|---|---|---|
| ELISA− | 4673 | 55.2 (13.6) | 0.57 | 0.54 | 25.6 (4.5) | 1.1 (1.4) | 5.8 (4.9) |
| ELISA±/IFT−/PRNT− | 33 | 58.5 (14.6) | 0.36 | 0.55 | 25.0 (3.9) | 1.1 (1.2) | 6.2 (5.1) |
| ELISA±/IFT+/PRNT− | 2 | 58.0 (8.5) | 1.00 | 0.00 | 27.1 (5.5) | 1.5 (0.7) | 3.5 (3.5) |
| ELISA±/IFT+/PRNT+ | 1 | 44 | 1 | – | 30.5 | – | 13 |
| ELISA+/IFT−/PRNT− | 23 | 54.6 (13.5) | 0.52 | 0.74 | 25.3 (5.4) | 1.0 (0.9) | 5.6 (3.6) |
| ELISA+/IFT+/PRNT− | 7 | 63.7 (20.9) | 0.57 | 0.43 | 28.3 (6.2) | 1.1 (1.1) | 5.6 (3.4) |
| ELISA+/IFT+/PRNT+ | 16 | 52.5 (15.4) | 0.69 | 0.62 | 23.6 (2.1) | 0.9 (0.9) | 9.5 (6.2) |
| Missing | 16 | 56.4 (9.7) | 0.69 | 0.50 | 28.6 (6.3) | 1.4 (1.3) | 4.2 (4.3) |

Values represent means (standard deviation) for continuous variables and fractions for categorical variables.
BMI body mass index, f female, IFT immunofluorescence test, PRNT plaque reduction neutralization test, y years.

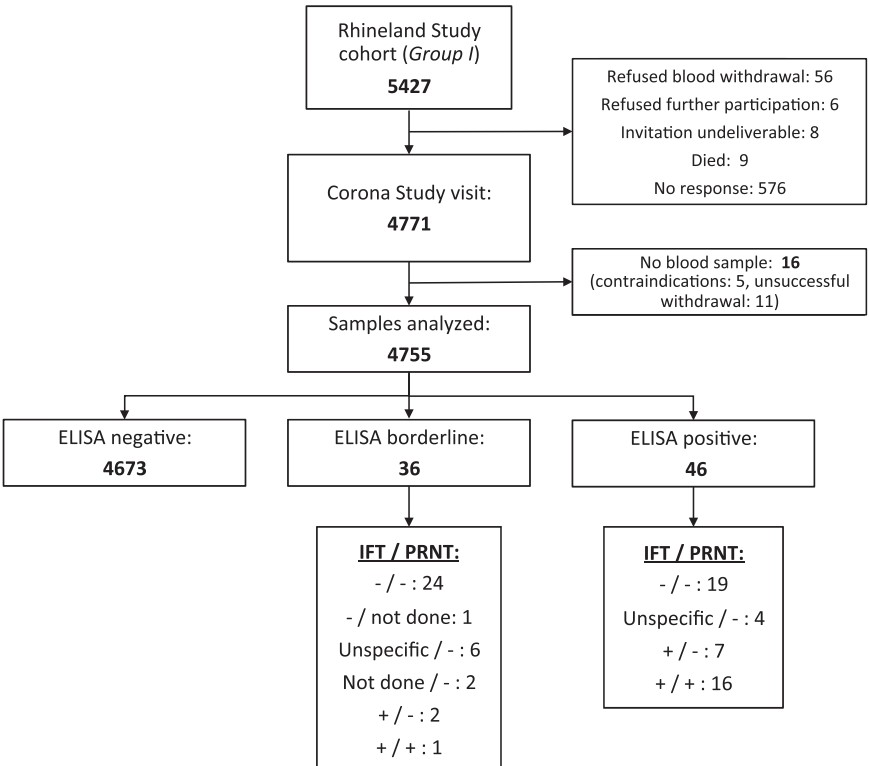

**Fig. 1 Flow chart.** Overview of the number of participants, their test results as well as reasons for non-participation or missingness. A minus or a plus sign indicates a negative or positive confirmatory test result, respectively. IFT immunofluorescence test, PRNT plaque reduction neutralization test.

thereby provides an estimate of the lower bound of the seroprevalence, the prevalence estimate was 0.36% (95% CI: 0.21–0.61). Thus, the true seroprevalence in Group I was estimated to lie between 0.36 and 0.97%.

The seroprevalence estimates were neither significantly associated with age (odds ratio (OR) 1.00 (95% CI: 0.98–1.02) for ELISA, and OR 0.98 (95% CI: 0.94–1.03) for all three tests) nor sex (male vs. female OR 0.94 (95% CI: 0.54–1.66) for ELISA, and 0.56 (95% CI: 0.23–1.40) for all three tests).

**Factors associated with the presence of neutralizing antibodies.** The 17 individuals with neutralizing antibodies came from 15 different households, including two households with two cases each and two households with one participant with and one without neutralizing antibodies. Only one individual had suffered from severe COVID-19 requiring hospitalization and intensive care treatment. The other 16 individuals had not required hospital care and were therefore assumed to have had asymptomatic or mild to moderate infection: All of them reported having experienced at least one symptom since January 1st 2020 (Supplementary Fig. 1), with the odds of having neutralizing antibodies increasing with 12% (OR 1.12, 95% CI: 1.04–1.21) for each additional symptom reported. Apart from living with a person with a confirmed or suspected SARS-CoV-2 infection and the number of experienced symptoms, other factors—including education, body mass index, comorbidity, alcohol consumption, smoking, and vaccination against seasonal influenza, pneumococcus or tuberculosis—were not associated with the presence of neutralizing antibodies (Fig. 2a); whereas a reduced sense of taste or smell, fever in the last month, chills or hot flashes, pain while breathing, pain in the arms or legs, as well as muscle pain and weakness were all significantly associated with the presence of neutralizing antibodies (ORs ranging from 3.44 to 9.97, all $p < 0.018$; Fig. 2b). Neither the total number of comorbidities nor

the presence of a particular comorbidity was associated with the presence of neutralizing antibodies (Fig. 2a and Supplementary Fig. 2).

**Factors associated with the presence of neutralizing antibodies in ELISA+ individuals.** In the subgroup of 46 ELISA+ individuals, those with neutralizing antibodies had a significantly higher antibody response as compared to those without neutralizing antibodies (age-adjusted and sex-adjusted difference in immunoglobulin G (IgG) ratio: 2.62, 95% CI: 1.81–3.43) (Fig. 3a). In addition, only in those with neutralizing antibodies the IgG response significantly increased with age (0.08 per year (95% CI: 0.05–0.12) in the ELISA+/PRNT+ subgroup, and 0.05 per year (95% CI: −0.002 to 0.10) in the ELISA+/PRNT− subgroup; Fig. 3b).

None of the 30 ELISA+/PRNT− individuals reported living in a household with a member with a previously confirmed SARS-CoV-2 infection, whereas three ELISA+/PRNT+ individuals indicated living together with a person with a previously confirmed SARS-CoV-2 infection (Fisher's exact test $P = 0.05$). Neither age (OR 0.98, 95% CI: 0.94–1.02) nor sex (male vs. female OR 0.47, 95% CI: 0.14–1.6) differentiated between the two groups. Those with neutralizing antibodies reported having experienced more symptoms (OR 1.19, 95% CI: 1.02–1.38), whereas other factors—including education, body mass index, comorbidity, alcohol consumption, smoking, and vaccination against seasonal influenza, pneumococcus, or tuberculosis—did not differentiate between the two groups (Fig. 4a). Fever (in the last month) and earache were only reported in the group with neutralizing antibodies by two and four individuals, respectively (Fisher's exact test $P$-values of 0.14 and 0.02 for fever and earache, respectively). The odds of neutralizing antibody seropositivity significantly increased with loss of appetite, muscle weakness, chills or hot flashes, and a reduced sense of taste (Fig. 4b).

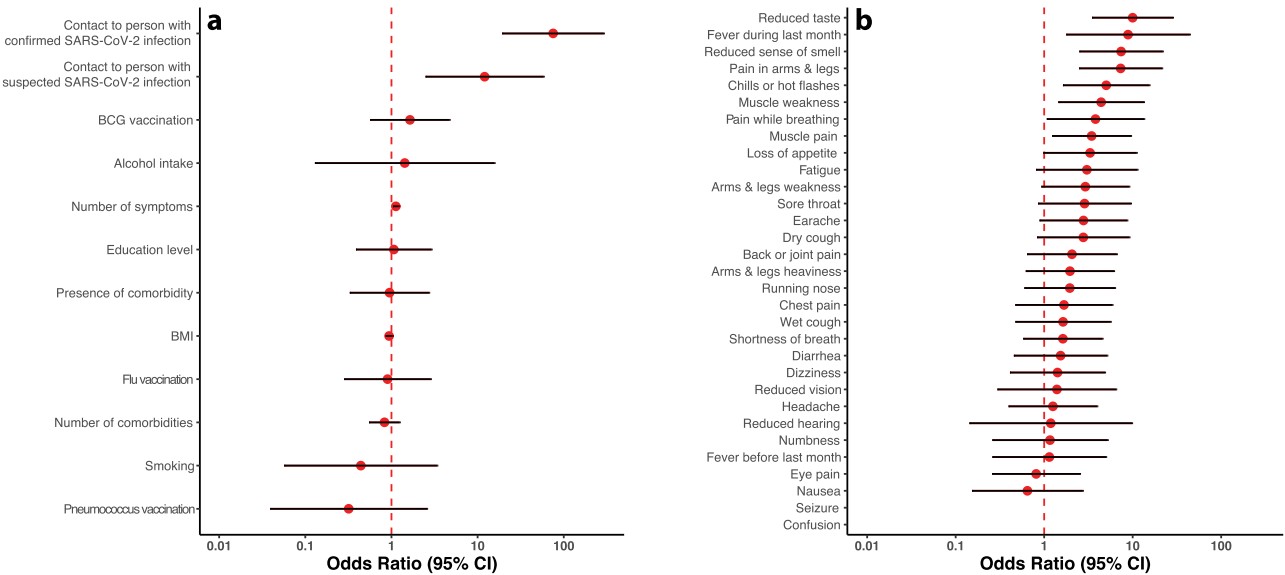

**Fig. 2 Factors associated with the presence of neutralizing antibodies. a** Living in the same household with a person with confirmed or suspected SARS-CoV-2 infection as well as a higher number of reported symptoms were significantly associated with the odds of having neutralizing antibodies. **b** A reduced sense of taste or smell, fever in the last month, pain in arm/legs, chills/hot flashes, pain while breathing as well as muscle weakness and pain were significantly more often reported by individuals with versus those without neutralizing antibodies; seizures and confusion were not reported in the seropositive group, and because of a very low background prevalence, the associated odds ratios could not be estimated reliably. The statistical comparisons were between 16 cases with mild-to-moderate symptoms who had SARS-CoV-2 neutralizing antibodies and 4754 individuals without SARS-CoV-2 (neutralizing) antibodies. All estimates are adjusted for age, sex and household clustering. The red dots represent the odds ratio point estimates, while the whiskers depict the corresponding 95% confidence intervals, on a logarithmic scale.

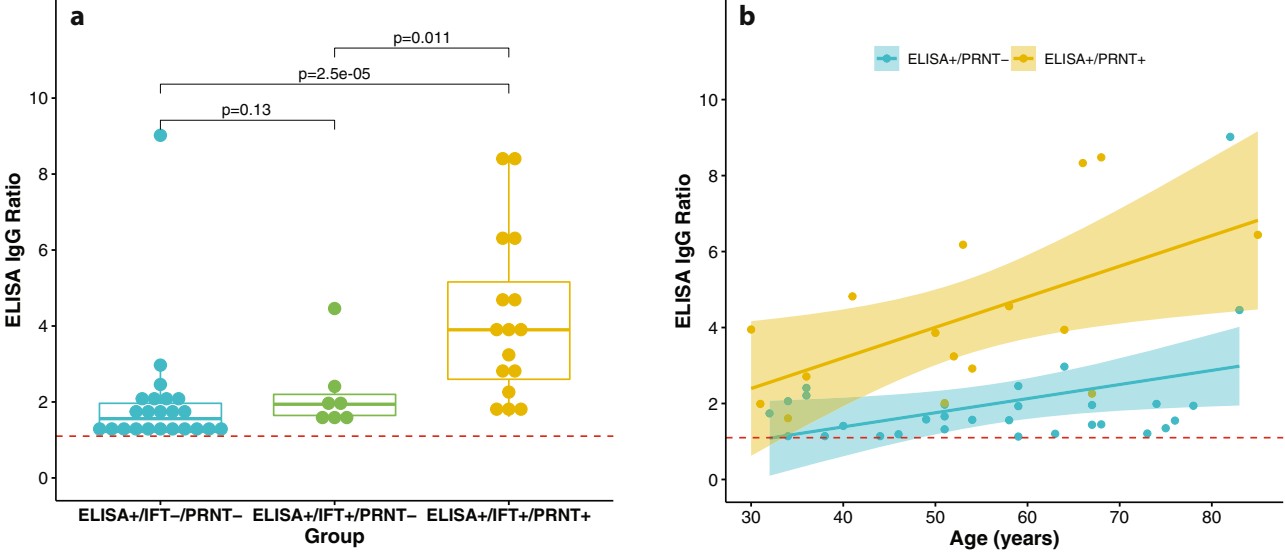

**Fig. 3 Relation between IgG response and neutralizing antibodies. a** Individuals with neutralizing antibodies had a significantly higher IgG antibody response as represented by the ELISA IgG ratio (a minus or a plus sign indicates a negative or positive test result, respectively). Each individual measurement is represented by one dot. The red dotted line indicates the threshold for a positive ELISA result. The box-plots indicate the medians (bold horizontal lines) and interquartile ranges (box boundaries), while the whiskers represent 1.5× interquartile ranges. Sample sizes: 23 ELISA+/IFT−/PRNT−, 7 ELISA+/IFT+/PRNT−, and 17 ELISA+/IFT+/PRNT+. *P*-values were obtained by the two-sided non-parametric Wilcoxon test. **b** Only in the ELISA+/PRNT+ subgroup there was a significantly higher IgG response with increasing age. The shaded areas around the regression lines represent the 95% confidence intervals of the mean. The red dotted line indicates the threshold for a positive ELISA result. Please refer to the main text for further details. IFT immunofluorescence test, PRNT plaque reduction neutralization test.

**Seroprevalence estimates in Group II**. A summary of the sample characteristics and test results of this group is presented in Supplementary Table 1. The seroprevalence in Group II was 1.94% (95% CI: 0.84–4.42) by ELISA and 1.39% (95% CI: 0.49–3.85) by all three tests (including one individual with a borderline positive immunofluorescence who had neutralizing antibodies). Thus, the

true seroprevalence in Group I was estimated to lie between 1.39 and 1.94%. Compared to Group I, the odds of a positive ELISA result were two-fold higher in Group II, although the OR did not reach statistical significance (OR 2.03, 95% CI: 0.82–4.99). The odds of having neutralizing antibodies were almost four-fold higher in Group II compared to Group I (OR 3.88, 95% CI:

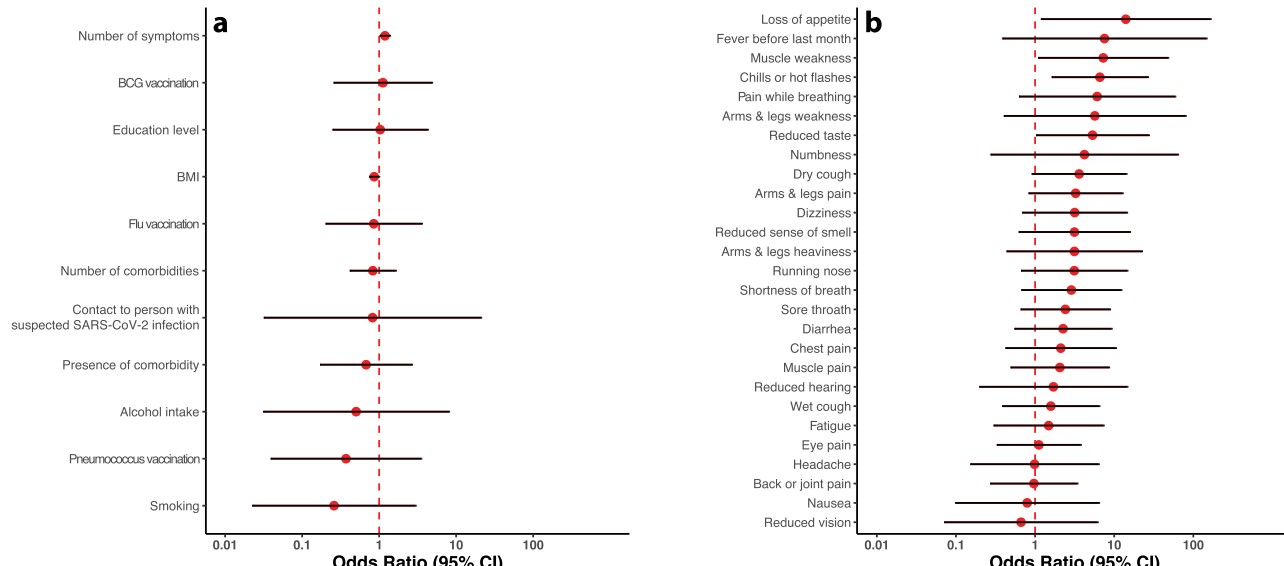

**Fig. 4 Factors differentiating between ELISA+ individuals with and without neutralizing antibodies. a** A higher number of reported symptoms was significantly associated with a higher odds of having neutralizing antibodies. **b** Loss of appetite, muscle weakness, chills or hot flashes as well as a reduced sense of taste were significantly more often reported by individuals with versus those without neutralizing antibodies. All estimates are adjusted for age, sex and household clustering. The statistical comparisons were between 16 ELISA+/PRNT+ cases and 30 ELISA+/PRNT− control subjects. The red dots represent the odds ratio point estimates, while the whiskers depict the corresponding 95% confidence intervals, on a logarithmic scale.

1.20–12.55). Groups I and II did not differ with respect to age, sex, and number of comorbidities (all $P \geq 0.07$). However, as compared to Group I, Group II individuals were slightly less likely to report symptoms (OR 0.97, 95% CI: 0.95–1.00). Importantly, however, the proportion of people who reported having a household member with a confirmed SARS-CoV-2 infection was substantially higher in Group II compared to Group I (1.67% vs. 0.36%, adjusted $P = 0.007$). In addition, a higher proportion of people in Group II, as compared to Group I, reported having been previously tested positive for a SARS-CoV-2 infection (1.11% vs. 0.23%, adjusted $P = 0.016$).

**Short-term follow-up of individuals with borderline ELISA results.** All individuals from both Group I and II, who had a borderline ELISA result at baseline were invited for a return visit within about 8 weeks of the first blood withdrawal. 30 of the 39 invited individuals returned for follow-up testing after a median of 28 days (range 20–41 days). At follow-up the IgG ratio had increased beyond 1.1 in seven of these individuals; however, neutralizing antibodies could not be detected in any of these participants at the follow-up visit, not even in the individual in whom the presence of neutralizing antibodies was confirmed at the baseline visit (Supplementary Fig. 3).

**Long-term follow-up of individuals with either borderline or positive ELISA results.** In September 2020, we re-invited all 92 individuals from both Group I and II, who had an IgG level in the borderline or positive range (i.e., ≥0.8) at baseline. Of these individuals, 83 came for a follow-up blood withdrawal (response rate ≈ 90%), including all 22 individuals in whom the presence of neutralizing antibodies had previously been confirmed. The blood withdrawals were performed after a median of 120 days from the baseline visit (range 89–144 days).

On average the levels of neutralizing antibodies declined over the follow-up period (OR for increase in $PRNT_{50}$ titer levels per day = 0.99, 95% CI: 0.98–0.99), with neutralizing antibodies becoming undetectable in 4 out of 22 individuals who were PRNT + at baseline (Fig. 5a). The IgG antibody response was relatively

modest in all four individuals in whom neutralizing antibodies could not be detected after long-term follow-up (Fig. 6). A higher IgG response over the follow-up period was inversely associated with the probability of becoming neutralizing antibody negative (OR for time (in days) × IgG ratio interaction = 0.96, 95% CI: 0.94–0.98, $P << 0.001$). Similarly, higher titers of neutralizing antibodies at baseline were associated with a lower probability of becoming neutralizing antibody negative after follow-up (OR for time (in days) × $PRNT_{50}$ titers interaction = 0.18, 95% CI: 0.18–0.19, $P << 0.001$). There was also a group of four individuals, with $PRNT_{50}$ titers in the >1:80 range at both visits, in whom the titers of neutralizing antibodies robustly increased during follow-up as evidence by higher $PRNT_{90}$ at the follow-up as compared to the baseline visit (Fig. 5b and Supplementary Table 2). The magnitude of the neutralizing antibody response over the follow-up period was positively associated with the presence and number of comorbidities, and inversely related to the body mass index (BMI) (Supplementary Fig. 4).

## Discussion

We present the findings of the largest population-based SARS-CoV-2 seroepidemiological cohort study to date in Germany. We conducted a study between April 24th and June 30th, 2020, in Bonn, a middle-large city in the western part of Germany, which at the time had a relatively low burden of COVID-19[25,26], and found a low prevalence of SARS-CoV-2 seropositivity. In addition, we found that: (1) only about one third of the individuals who tested positive on a widely used quantitative immunoassay had detectable levels of serum neutralizing antibodies; (2) both the magnitude of the antibody response, as reflected by the IgG ratio, the total number of symptoms experienced, as well as the presence of particular symptoms were associated with the presence of neutralizing antibodies in those with a positive immunoassay test result; (3) in those with a borderline immunoassay result the presence of neutralizing antibodies was extremely rare, and—in the only confirmed case—transient, (4) the titers of neutralizing antibodies wane relatively quickly, decreasing below detection limit in a sizeable proportion (about 20%) within a few

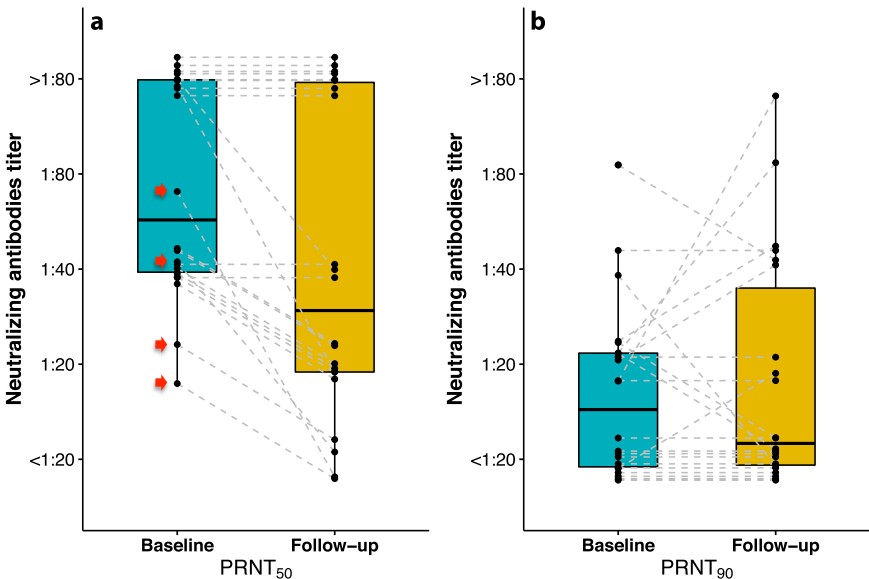

**Fig. 5 Time course of neutralizing antibody response. a** Titers of neutralizing antibodies decreased in most individuals during follow-up, becoming undetectable in four (PRNT$_{50}$ titers; the four individuals in whom neutralizing antibodies became undetectable after follow-up are highlighted by red arrows). **b** Within the subgroup of individuals with the highest neutralizing antibody titers at baseline, there were four individuals in whom these titers continued to increase (PRNT$_{90}$ titers). Each two-points joined by a dashed line represent one participant. The box-plots indicate the medians (bold horizontal lines) and interquartile ranges (box boundaries), while the whiskers represent 1.5× interquartile ranges. Sample size: 22 individuals with SARS-CoV-2 neutralizing antibodies at baseline. Note that neutralizing antibody titers were measured on a semi-quantitative scale (i.e., <1:20 (undetectable), 1:20, 1:40, 1:80, or >1:80), therefore, random vertical jitter was added to the points to avoid masking of data points with similar values. PRNT plaque reduction neutralization test.

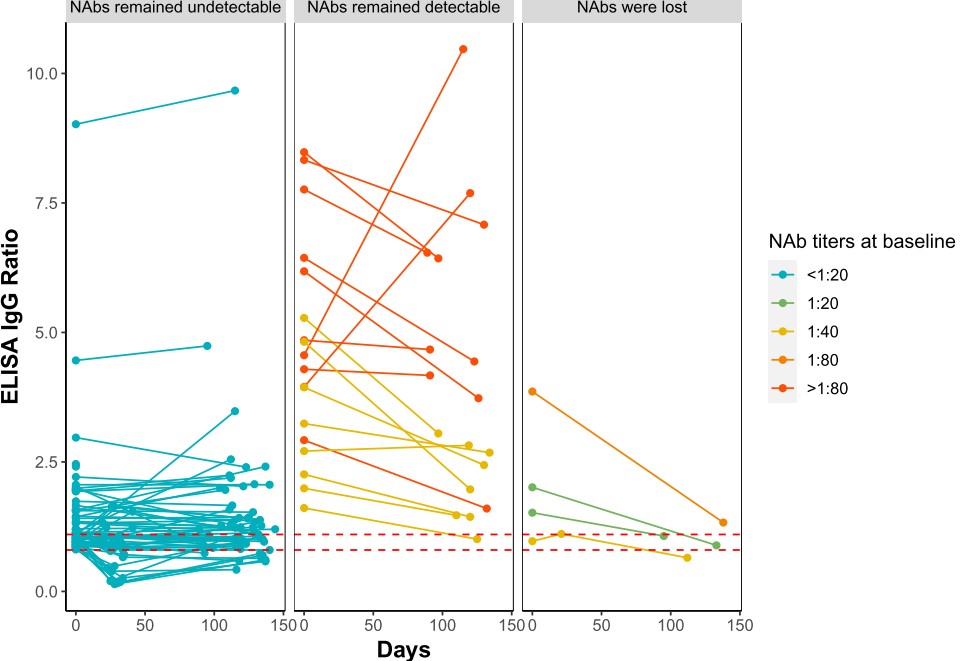

**Fig. 6 Time course of immunoassay-based IgG levels in relation to neutralizing antibody response at baseline.** There was a strong association between the magnitude of the immunoassay-based IgG response and the probability of levels of neutralizing antibodies (NAb) decreasing below detection limit (odds ratio for time (in days) × IgG ratio interaction = 0.96, 95% CI: 0.94–0.98, two-sided $P << 0.001$, using generalized estimating equations with a cumulative logistic link function and an independent covariance structure, and deriving the confidence intervals from the robust Huber-White sandwich variance estimator). The four individuals, who became NAb negative during follow-up, had relatively modest IgG responses (right panel). Sample size: 22 individuals with SARS-CoV-2 neutralizing antibodies at baseline. The horizontal dotted lines represent the borders of the indeterminate range.

months, with the probability of neutralizing antibody loss being inversely related to the magnitude of the IgG response, and (5) self-referral bias can lead to substantial overestimation of seroprevalence.

As of June 30th, there were a total of 759 confirmed SARS-CoV-2 infections (including three COVID-19 related deaths) in Bonn, a city with about 330,000 inhabitants, yielding a prevalence of 0.23%[29]. Our seroprevalence estimates are 1.6–4.2 times higher (based on confirmation by all three tests or a positive ELISA result only, respectively). The study sample did not include people younger than 30 years old and is, therefore, not representative of the entire Bonn population. Nevertheless, our findings suggest that a considerable number of individuals who had been infected went undetected by the local health regulatory agencies, most likely because of subclinical infection or development of mild symptoms only. Our findings also confirm that limited exposure of the local population to SARS-CoV-2 most likely accounts for the relatively low rates of regional COVID-19-related hospitalizations and mortality at the time, supporting the efficacy of early implementation of social distancing and confinement measures in Germany[25,26]. Our seroprevalence estimates are lower than the three other German community-based serosurveys, which were conducted during the first wave of the pandemic[22–24]. However, these previous serosurveys were conducted in communities following super-spreading events, and are unlikely to reflect the state of other regions in Germany with relatively low COVID-19 burden[22–24].

Neutralizing antibodies were detected in only about one third of the participants who tested positive on a widely used immunoassay. At least two explanations may account for this finding. First, the individuals who tested positive on the immunoassay but not on the confirmatory tests, may be false positives, e.g., due to cross-reactivity with antibodies against other coronaviruses. Based on a specificity of 99.6% for the ELISA that we used and assuming a zero prevalence, we would have expected 19 false positives[27]. Second, this group may also include individuals who were infected with SARS-CoV-2, but who either did not develop neutralizing antibodies or lost them in the period following infection. Indeed, we could not detect neutralizing antibodies in 6 out of 15 people who reported to have had a SARS-CoV-2 infection in the past. In addition, we found that neutralizing antibody titers decreased below detection in about 20% of seropositive individuals within just 5 months of follow-up. Although another recent study found a more sustained neutralizing antibody response, the study used a microneutralization assay and the cohort included a substantial group of people who were referred for screening based on suspected COVID-19, and thus likely had more severe disease, not representing an unbiased population-based sample[30]. Conversely, our findings further extend other recent reports indicating that neutralizing antibodies may not develop in asymptomatic or only mildly symptomatic individuals and, especially in this group, may wane relatively quickly in the period following infection[28]. Although neutralizing antibodies are thought to be a major component of adaptive immunity and their levels also strongly correlate with the number of SARS-CoV-2 specific T-cells[31], a decline of neutralizing antibody levels does not necessarily imply absence of persisting immune memory[32]. Indeed, SARS-CoV-2 specific T-cells recognizing peptides derived from the spike, nucleoprotein, and matrix components of the virus have also been found in convalescent plasma following mild COVID-19 and could potentially confer a longer lasting immunity[33]. Similarly, despite decreasing antibody levels after about three weeks post-symptom onset, a recent study found that the number of long-lived memory B cells in blood samples obtained from 25 convalescent COVID-19 patients continued to rise until about 5 months[34]. Though there is still much to be

learned about which components of the immune response provide protection against both initial infection and reinfection, recent evidence from SARS-CoV-2 vaccine trials support the notion that the presence and magnitude of neutralizing antibodies can be regarded as a valid marker of a protective immune response[35–37]. Therefore, the possibility cannot be excluded that even after infection a large proportion of individuals in the general population, especially those with asymptomatic or mild infections, may become susceptible again to SARS-CoV-2 infection.

Among persons with a positive immunoassay result, the magnitude of the IgG response, the number of previously experienced symptoms, as well as the prior occurrence of particular symptoms—including loss of appetite, muscle weakness, chills or hot flushes and reduced taste—were strongly associated with the probability of having neutralizing antibodies. In addition, titers of neutralizing antibody were also strongly associated with the magnitude of the IgG response, as well as some clinical features like the number of co-morbidities and BMI. These findings thus suggest that predictive models could be developed, based on a combination of clinical characteristics and immunoassay-based antibody levels to estimate both the probability and the magnitude of the neutralizing antibody response. Testing for neutralizing antibodies is currently very labor intensive and can only be reliably performed in specialized laboratories. Good prediction models could be useful to better estimate the actual population immunity level in regions without access to such advanced testing facilities.

Self-referral or volunteer bias could inflate seroprevalence estimates[38]. In order to estimate the magnitude of this effect, we also sampled a group of spontaneous volunteers from the same region who were not part of the original Rhineland Study cohort, but who expressed interest in the serosurvey. After formal invitation of these individuals, the response rate was almost 30% lower compared to the original cohort of participants, whereas the seroprevalence estimates were two to four-fold higher (based on the presence of neutralizing antibodies or a positive ELISA result, respectively). It appeared that the main reasons for self-selection were not so much the presence of symptoms, but a previously confirmed SARS-CoV-2 infection or the presence of a close contact with a previous SARS-CoV-2 infection. These findings thus illustrate the profound impact of selection-bias on seroprevalence estimates—likely a major source for the large heterogeneity of the findings of many previous serosurveys—and thereby underscore the critical importance of cohort-based analyses that allow for accurate quantification of response rate and reasons for (non-)response.

Our study has both strengths and limitations. By implementing this serosurvey in an ongoing community-based prospective cohort study, we were able to quickly reach and mobilize a large group of participants and achieve a very high response rate, thereby minimizing the risk of selection bias. Indeed, to the best of our knowledge, our serosurvey is the only available study in which the magnitude of self-selection bias has been specifically addressed with respect to SARS-CoV-2 seroprevalence. By presenting differences in the seropositivity rates for participants from Group I (low risk of self-selection) and Group II (high risk of self-selection), who otherwise originated from the same pre-defined geographical region, our study can be used to illustrate and warn against the substantial influence of self-referral/self-selection bias in inflating seroprevalence estimates. We also present one of the very few follow-up studies targeting both the correlates and temporal evolution of the neutralizing antibody response at population-level. The limitations of our study include that it was restricted to adults ≥30 years, and the fact that we did not assess SARS-CoV-2 specific T-cell and memory B cell responses, other

critical components of acquired immunity[39,40]. Nevertheless, given the low (neutralizing antibody) seroprevalence, it is highly unlikely that the population seroprevalence in this region at the time would have materially differed from our estimates.

In conclusion, neutralizing antibodies to SARS-CoV-2 were detectable in only one third of those with a positive immunoassay result and waned relatively rapidly within a few months. These findings not only indicate that almost the entire population in this region was susceptible to SARS-CoV-2 infection at the time, but also suggest that without vaccination sustained population immunity to SARS-CoV-2 is difficult to achieve, warranting continued vigilance and implementation of countermeasures to curb the further spread of the infection by health authorities.

## Methods

**Study population**. This serosurvey was based on the Rhineland Study, an ongoing community-based cohort study in Bonn, Germany. All inhabitants aged 30 years and above of two geographically defined areas are invited to participate in the Rhineland Study. The sole exclusion criterion is insufficient command of the German language to provide informed consent. Persons living in the recruitment areas are predominantly German from Caucasian descent. The Rhineland Study's overarching aims are to investigate the etiology and prediction of age-related (neurodegenerative) diseases, and to assess normal and pathological (brain) structure and function over the adult life course[41]. The study started in 2016 and emphasizes deep phenotyping. Because of the imposition of local lockdown measures, regular study visits were paused on March 18th, 2020.

This serosurvey was conducted in two groups. Group I consisted of all living participants who had been enrolled in the Rhineland Study until March 18th, 2020 ($N = 5427$). Their participation in the study was therefore unrelated to attitude to or experience with SARS-CoV-2. Group II consisted of individuals who were eligible for but had not yet participated in the Rhineland Study. They actively approached us to indicate their willingness to participate in the serosurvey ($N = 597$), which we allowed for those who agreed to become prospective participants in the Rhineland Study. Participation in this group was thus motivated by the prospect of being tested.

Approval to undertake the Rhineland Study was obtained from the ethics committee of the University of Bonn, Medical Faculty (reference ID: 338/15). The Rhineland Study is carried out in accordance with the recommendations of the International Conference on Harmonization (ICH) Good Clinical Practice (GCP) standards (ICH-GCP) after obtainment of written informed consent from all participants in accordance with the Declaration of Helsinki. No separate ethical approval for this serosurvey was required given its embedding in the Rhineland Study, the ethical mandate of which already covered follow-up measurements, including collection of serial bio-samples.

**Study design and procedures**. All Group I participants were informed about the serosurvey through email, postal letter and/or phone, whereas, as stated above, Group II individuals actively approached us. All invitees were requested to use a purpose-designed online platform to make an appointment at one of the two local study centers, except when they were suffering from symptoms of an acute infection (especially fever, cough or other flu-like complaints), in which case they were recommended to visit a doctor. From April 24th through June 30th, 4771 (88%) of the invited participants from Group I and 360 (60%) of the invited participants from Group II, visited one of the study centers for a blood withdrawal. Reasons for non-response included death, undeliverable invitations, and refusal due to a perceived high burden or risk of infection due to old age, immobility, or co-morbidity.

**Data collection**. Data collection included blood withdrawal and questionnaires. At the study center, blood was collected from an antecubital or dorsal hand vein. 30–90 min after blood withdrawal serum tubes were centrifuged for 15 min at $2000 \times g$, stored directly at +2 to +8 °C thereafter, and sent to the diagnostic lab via overnight courier within about 1 week of collection. Following blood collection, participants received a paper questionnaire that they were asked to complete at home and return to the study center.

**Serological measurements**. Serological analyses for SARS-CoV-2 antibodies were performed at the National Consultant Laboratory for Coronaviruses (Institute of Virology, Charité—Universitätsmedizin Berlin, Germany) using a three-tiered approach. First, the levels of IgG antibodies against SARS-CoV-2 were measured using a commercially available ELISA (EUROIMMUN, Lübeck, Germany). According to the manufacture's product sheet, applying a cut-off of >1.1 for defining seropositivity results in an estimated sensitivity of 94.4% (at >10 days of infection) and specificity of 99.6%. Our independent in-house validation experiments of this assay using serum samples from 119 plasma donors after convalescence from mild to moderate COVID-19 (as documented by positive swab

nucleic acid testing), as well as 110 healthy subjects (with either no history of COVID-19-typical symptoms and no risk contacts, or negative pharyngeal swab SARS-CoV-2 nucleic acid testing), were in line with the manufacturer's reported values yielding a sensitivity of 86.8% and a specificity of 100%[42]. Importantly, we re-invited all those individuals with IgG levels ≥0.8 at baseline for repeated follow-up measurements of the IgG levels within 3–6 weeks so as not to miss individuals with a nascent humoral immune response after a recent infection. We performed two additional confirmatory tests in all those individuals whose ELISA assay results were either positive (i.e., >1.1) or borderline (i.e., between 0.8 and 1.1). Confirmatory tests consisted of an in-house recombinant immunofluorescence test and a PRNT to specifically check for the presence of neutralizing antibodies against SARS-CoV-2 as we described recently[43,44]. Neutralizing antibody titers were measured as the concentration of serum to reduce the number of plaques by either 50 or 90% (defined as PRNT$_{50}$ or PRNT$_{90}$, respectively) and categorized as <1:20 (i.e., undetectable), 1:20, 1:40, 1:80 or >1:80. Especially in regions where seroprevalence is expected to be low, minimizing the number of false positives is crucial. Therefore, we tested the specificity of our three-tiered testing approach, using 100 randomly selected pre-pandemic samples from the same cohort (i.e., Group I), which were collected between 2016 and 2018 during the months March to September to roughly correspond to the timing of sample collection during the pandemic in our serosurvey. All 100 samples tested negative on the commercial ELISA, the in-house recombinant spike-based immunofluorescence test and PRNT, confirming the extremely high specificity of our three-tiered testing approach (Supplementary Table 3 and Supplementary Fig. 5).

**Questionnaires**. Data on physical and mental health were collected through an extensive questionnaire addressing current demographic, living and socioeconomic conditions, co-morbidities, medication and substance use, as well as COVID-19 related symptoms. The questions were selected taking account of other ongoing and developing COVID-19 related studies (especially the various "COVID-19 Host Genetics Initiative" cohorts[45], particularly the Lifelines study;[46] https://www.covid19hg.org), to facilitate future data harmonization, sharing, and collaboration).

**Statistical analysis**. Descriptive statistics are presented as means and 95% CIs for continuous variables or numbers and percentages for categorical variables. Generalized estimating equations (GEE) with an independent covariance structure within household units were used to account for potential correlations between members of the same household. We used GEE with a logistic link function to estimate seroprevalence. We also applied GEE with either a logistic or Gaussian link function to assess which factors were associated with seropositivity or IgG ratio, respectively, while adjusting for potential confounders. We specifically assessed whether age, sex, education (as a measure of socioeconomic status), pre-existing medical conditions, vaccination, BMI, smoking, or alcohol consumption were associated with these outcomes. In addition, we assessed the relation between the number of different symptoms experienced since January 1st, 2020 (regardless of whether occurring at the same time or at different times), until the time of blood withdrawal and seropositivity. GEE models with a cumulative logistic link function and an independent covariance structure to account for repeated intra-individual follow-up measurement were used to assess the temporal dynamics and correlates of the neutralizing antibody response[47]. All GEE confidence intervals were based on the robust Huber-White sandwich variance estimator. In case of subgroups with zero counts, Fisher's exact test was used instead for intergroup comparisons. All analyses were performed in R (base version 3.6.1). The following R packages were used for the statistical analyses: geepack (version 1.3-1), multgee (version 1.7.0) and stats (version 3.6.3). A two-tailed P-value of <0.05 was considered statistically significant.

**Reporting summary**. Further information on research design is available in the Nature Research Reporting Summary linked to this article.

## Data availability

The Rhineland Study's dataset is not publicly available because of data protection regulations. Access to data can be provided to scientists in accordance with the Rhineland Study's Data Use and Access Policy. Requests for further information or to access the Rhineland Study's dataset should be directed to RS-DUAC@dzne.de. Source data are provided with this paper.

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

## Acknowledgements

The authors would like to thank all the staff members and participants of the Rhineland Study. We would especially like to thank Theda Backen, Nüket Cebi, Karin Heubach, Angelika Klein, Stefan Kurth and Ina Plettner for their dedicated support in the planning and management of this research project; Nersi Alaeddin, Felix Ernst, Elvire Landstra, Dan Liu, Marina Santos, Valentina Talevi, and Weiyi Zeng for valuable assistance with the processing of the large number of bio-samples; and Simon Harmata and André Medek for relentless IT-support. We also thank Elisabeth Möncke-Buchner and Nicolas Heinemann (both from Charité) for technical assistance. The Rhineland Study is predominantly funded through the German Center for Neurodegenerative Diseases (DZNE) by the Federal Ministry of Education and Research (BMBF) and the Ministry of Culture and Science of the German State of North Rhine-Westphalia. This work was also supported by the Deutsche Forschungsgemeinschaft (DFG, German Research Foundation) under Germany's Excellence Strategy – EXC 2151 – 390873048. The National Consultant Laboratory for Coronaviruses is funded by the Federal Ministry of Health (BMG). The funders were neither involved in study design, data collection, analysis, and interpretation, nor in writing of the paper and the decision to submit for publication. No additional funding was received for this seroprevalence study.

## Author contributions

N.A.A., V.M.C., A.S., C.D., and M.M.B.B. conceived the study. N.A.A., V.M.C., M.A.M., F.M.dV., A.K.C.E., T.H.S., A.S., C.D., and M.M.B.B. participated in the study design. F.M. dV., A.K.C.E., T.H.S., and A.S. coordinated the data and bio-sample collection and processing. V.M.C., M.A.M., M.L.S., A.R., and C.D. supervised and coordinated the performance of the serological assays. N.A.A. did the data analyses. N.A.A. and M.M.B.B. drafted the first version of the manuscript. All authors contributed to data interpretation, participated in revising the manuscript for important intellectual content, and read and approved the final manuscript.

## Funding

## Competing interests

Dr. Marcel A. Müller and Dr. Victor M. Corman are named together with EURO-IMMUN on a patent application filed recently regarding antibody diagnostics of SARS-CoV-2. All the other authors declare no competing interests.
