## [Peer Review File · Nature Communications]

Reviewers' comments:

Reviewer #1 (Remarks to the Author):

In their manuscript Aziz and colleagues determine the seroprevalence against SARS-CoV-2 in Bonn, Germany. The manuscript is interesting but there are several points that require the authors' attention.

Major points

1) All seroprevalence studies in areas with low seroprevalence and assays that are not close to 100% in specificity are in a way problematic. Here, 0.4% false positives are expected, a significant part of the positives detected. The authors perform three assays to determine positivity and the vast majority only tests positive in one assays – in fact the same assay. Those samples are problematic because it is completely unclear if they are true positives or not. And if they are not, the interpretation of the manuscript changes dramatically.

2) Two of the assays have no given specificity or sensitivity. For the third assay the performance metrics from the manufacturer are used, which are usually better than their field metrics. Since this study is embedded in another long term study, the authors should test their three assay with at least 100 prepandemic samples plus a good number of samples from confirmed PCR positive individuals. In house determined sensitivity and specificity would strengthen the dataset considerably.

3) Why do several samples go up significantly for the PRNT90 but not for the PRNT50? This seems odd.

4) Please describe how the group (group 2) that 'expressed interest' and was enrolled could bias the study.

Minor points

1) Many abbreviations are not defined.

2) Line 91: This sentence has a weird grammar structure.

3) Line 376-380: Consider memory B cells, especially due to the long incubation time.

Reviewer #2 (Remarks to the Author):

In this interesting study the authors undertake a community study and assess neutralising antibody status.

The work is well performed with no technical concerns.

The second cohort is somewhat less representative, smaller and likely to include many people who agreed to take part as they had symptoms.

The statistical analysis is somewhat weakened by the small number of donors with neutralising antibodies.

This is a highly competitive area but the paper adds value in relation to presence of neutralising antibodies

-over time

- in relation to total antibody response

-symptoms

Reviewer #3 (Remarks to the Author):

This is a well done study on an important area that is reported well.

I just have a few comments about the presentation and interpretation of the results.

My main comment refers to the interpretation of these results for herd immunity. I would like to see a greater discussion of T-cell immunity and the uncertainties about what is measured for antibody and T cell immunity and protection.

Minor comments

Line 201: Is this number of symptom episodes or numbers of different symptoms described regardless of whether at same or different times? Please clarify.

Line 225: I would suggest seroprevalence instead of prevalence as prevalence can be very easily confused with the prevalence of current infection.

Line 321: Suggest removing "interestingly", not that it isn't interesting, but I don't think it is more interesting than the declines, given the study probably captured people at different times after infection, an increase in some seems reasonable to me.

Line 331: Please restate the time period for the seroprevalence estimates here.

Line 350: I'm not sure about the use of the word "evaded" this is to be expected given the asymptomatic nature of many infections. Perhaps this could be included in the discussion here.

Line 354: Suggest use of date rather than "current".

Line 376: I would suggest a greater discussion of T cell immunity, specifically here I think the correlation between antibody and T cell is still unclear. Generally I think the statement here and in the abstract about herd immunity is still too strong given these uncertainties

POINT-BY-POINT RESPONSE TO THE REVIEWERS' COMMENTS:

REVIEWER #1:

In their manuscript Aziz and colleagues determine the seroprevalence against SARS-CoV-2 in Bonn, Germany. The manuscript is interesting but there are several points that require the authors' attention.

Major point 1:

"All seroprevalence studies in areas with low seroprevalence and assays that are not close to 100% in specificity are in a way problematic. Here, 0.4% false positives are expected, a significant part of the positives detected. The authors perform three assays to determine positivity and the vast majority only tests positive in one assays – in fact the same assay. Those samples are problematic because it is completely unclear if they are true positives or not. And if they are not, the interpretation of the manuscript changes dramatically."

Reponse:

As reported both in the abstract and in the body of the manuscript, we based the SARS-CoV-2 seroprevalence estimates on the results of three different tests, i.e. an ELISA, followed by confirmatory testing of borderline and positive test results with a recombinant spike-based immunofluorescence assay and a plaque reduction neutralisation test (PRNT). We distinctively report the prevalence estimates using two different approaches: 1) The prevalence estimates based on ELISA results only, an approach which prioritizes *sensitivity* and thereby provides an estimate of the upper-bound of the seroprevalence estimate, and 2) The prevalence estimates based on the results of the confirmatory tests, i.e. only treating those participants as cases who had tested positive on *all three tests* (all participants who had a positive PRNT test result, also had positive immunofluorescence and ELISA results (except one individual who had borderline ELISA results, but tested positive on both the immunofluorescence assay and PRNT, Figure 1 in the manuscript); an approach that prioritizes *specificity* and thereby provides an estimate of the lower-bound of the seroprevalence estimate. By reporting the findings based on these two complementary approaches, we in fact provide highly accurate estimates of the true seroprevalence in the source population, which thus is likely to lie between 0.36 to 0.97%. Therefore, the reviewer's characterization that "... the vast majority only tests positive in one assay – in fact the same assay..." appears to be based on a misunderstanding, which may stem from the way we reported our findings.

In order to convey our approach and findings more clearly in this regard, both in the abstract and discussion section of the manuscript, we now explicitly state the distinction between the prevalence estimates based on the ELISA only, as well as the prevalence estimate after confirmation by at least two additional confirmatory tests (i.e. a positive result on the recombinant immunofluorescence assay and the presence of neutralizing antibodies).

We have adapted the abstract as follows:

"Seroprevalence was 0.97% (95% CI: 0.72–1.30) by ELISA and 0.36% (95% CI: 0.21–0.61) when restricting cases to only those with two additional positive confirmatory tests."

In addition, we have added the following to the second paragraph of the results section, under the subheading "Prevalence estimate" of the manuscript:

*“The prevalence estimate based on ELISA results only, an approach that prioritizes sensitivity and thereby provides an estimate of the upper bound of the seroprevalence, was 0.97% (95% CI: 0.72–1.30). When only treating those participants as cases who had tested positive in all three tests (including one individual who had a borderline ELISA test result, but tested positive on both the recombinant immunofluorescence test and PRNT, **Figure 1**), an approach that prioritizes specificity and thereby provides an estimate of the lower bound of the seroprevalence, the prevalence estimate was 0.36% (95% CI: 0.21–0.61). Thus, the true seroprevalence in Group I was estimated to lie between 0.36 to 0.97%.”*

Major point 2:

“Two of the assays have no given specificity or sensitivity. For the third assay the performance metrics from the manufacturer are used, which are usually better than their field metrics. Since this study is embedded in another long term study, the authors should test their three assay with at least 100 prepandemic samples plus a good number of samples from confirmed PCR positive individuals. In house determined sensitivity and specificity would strengthen the dataset considerably.”

Response:

We agree with the reviewer that in-house validation of the assay characteristics would have been more rigorous and could further strengthen our findings. Indeed, we have now completed validation experiments of our in-house assay set-ups. For these validation experiments we used 100 randomly selected pre-pandemic samples from the same cohort (i.e. Group I), which were collected between 2016 and 2018 during the months March to September to roughly correspond to the timing of collection of the samples that were taken during the pandemic in our serosurvey. All 100 samples tested negative on the EUROIMMUN ELISA (using the same cut-offs as applied in our serosurvey), the in-house recombinant spike-based immunofluorescence test and the plaque reduction neutralisation test, confirming the extremely high specificity of our three-tiered test approach. We have included these findings as supplementary material in the paper (**Supplementary Table 1** and **Supplementary Figure 1**).

In addition, we recently published the results of extensive independent in-house validation experiments for the EUROIMMUN ELISA using serum samples from 119 plasma donors after convalescence from mild to moderate COVID-19 as documented by positive swab nucleic acid testing, as well as 110 healthy subjects with either no history of COVID-19-typical symptoms and no risk contacts, or negative pharyngeal swab SARS-CoV2 nucleic acid testing (Jahrsdörfer, Kroschel et al. 2020). Our validation results were in line with the company reported values and, for the EUROIMMUN anti-spike (S) IgG assay that we used with a cut-off of 1.1 to define a positive ELISA test result in our serosurvey, yielded a sensitivity of 86.8% and a specificity of 100%. Please note that given that the recombinant immunofluorescence and the PRNT tests were used as confirmatory tests where test specificity is most important, we did not assess the sensitivity of these confirmatory tests.

We have now adapted the methods section of the manuscript to include the following under the subheading “Serological measurements”:

“Serological measurements

*Serological analyses for SARS-CoV-2 antibodies were performed at the National Consultant Laboratory for Coronaviruses (Institute of Virology, Charité – Universitätsmedizin Berlin, Germany) using a three-tiered approach. First, the levels of immunoglobulin G (IgG) antibodies against SARS-CoV-2 were measured using a commercially available ELISA (EUROIMMUN, Lübeck, Germany). According to the manufacturer’s product sheet, applying a cut-off of >1.1 for defining seropositivity results in an estimated sensitivity of 94.4% (at >10 days of infection) and specificity of 99.6%. Our independent in-house validation experiments of this assay using serum samples from 119 plasma donors after convalescence from mild to moderate COVID-19 (as documented by positive swab nucleic acid testing), as well as 110 healthy subjects (with either no history of COVID-19-typical symptoms and no risk contacts, or negative pharyngeal swab SARS-CoV-2 nucleic acid testing), were in line with the manufacturer’s reported values yielding a sensitivity of 86.8% and a specificity of 100% (Jahrsdörfer, Kroschel et al. 2020). Importantly, we re-invited all those individuals with IgG levels ≥ 0.8 at baseline for repeated follow-up measurements of the IgG levels within 3-6 weeks so as not to miss individuals with a nascent humoral immune response after a recent infection. We performed two additional confirmatory tests in all those individuals whose ELISA assay results were either positive (i.e. >1.1) or borderline (i.e. between 0.8 and 1.1). Confirmatory tests consisted of an in-house recombinant immunofluorescent test and a plaque reduction neutralisation test (PRNT) to specifically check for the presence of neutralizing antibodies against SARS-CoV-2 as we described recently (Huang, Garcia-Carreras et al. 2020, Wölfel, Corman et al. 2020) Neutralizing antibody titers were measured as the concentration of serum to reduce the number of plaques by either 50 or 90% (defined as PRNT₅₀ or PRNT₉₀, respectively) and categorized as <1:20 (i.e. undetectable), 1:20, 1:40, 1:80 or >1:80. Especially in regions where seroprevalence is expected to be low, minimizing the number of false positives is crucial. Therefore, we tested the specificity of our three-tiered testing approach, using 100 randomly selected pre-pandemic samples from the same cohort (i.e. Group I), which were collected between 2016 and 2018 during the months March to September to roughly correspond to the timing of sample collection during the pandemic in our serosurvey. All 100 samples tested negative on the commercial ELISA, the in-house recombinant spike-based immunofluorescence test and PRNT, confirming the extremely high specificity of our three-tiered testing approach (**Supplementary Table 1 and Supplementary Figure 1**).”*

Major point 3:

“Why do several samples go up significantly for the PRNT90 but not for the PRNT50? This seems odd. “

Response:

This apparent discrepancy is due to the semi-quantitative nature of the plaque reduction neutralization tests (PRNTs). The finding that some samples go up for PRNT90 and not for PRNT50 is due to a ceiling effect of the scale on which these measurements are quantified: The highest category for both PRNT50 and PRNT90 is defined as “titer > 1:80” (Figure 5 in the manuscript). Therefore, even if the neutralizing antibody titer rises, this cannot be picked up in cases where the original PRNT50 category was already “titer > 1:80”. However, in these cases,

further rises in neutralizing antibody titers can be detected based on PRNT₉₀, i.e. the concentration of serum to reduce the number of plaques by 90%. Four of the five individuals with a clear rise in PRNT₉₀ levels were already in the highest PRNT₅₀ category. Only one individual with a borderline increase of PRNT₉₀ levels (i.e. from the <1:20 to the 1:20 category) had PRNT₅₀ titers of 1:40 at both visits; this latter case is likely due to inter-assay variability in quantifying the neutralizing antibody titers. Therefore, measuring both PRNT₅₀ and PRNT₉₀ values is in fact a further methodological strength of our study and a useful illustration of why it is important to measure and report both PRNT₅₀ and PRNT₉₀ values.

We have now clarified this point more explicitly in the manuscript by adapting the description in the last paragraph of the results section as follows (including a table (**Supplementary Table 3**) showing aligned data underlying each participant's PRNT₅₀ and PRNT₉₀ levels):

*"... There was also a group of four individuals, with PRNT₅₀ titers in the >1:80 range at both visits, in whom the titers of neutralizing antibodies robustly increased during follow-up as evidence by higher PRNT₉₀ at the follow-up as compared to the baseline visit (**Figure 5B, Supplementary Table 3**)."*

Major point 4:

"Please describe how the group (group 2) that 'expressed interest' and was enrolled could bias the study. "

Response:

Group II did not bias our study because we analyzed Group II separately from Group I for calculating seroprevalence estimates. Rather, we included Group II to illustrate how seroprevalence studies that only start recruiting during the pandemic, are likely biased due to self-selection of participants.

The strength of our seroprevalence study is that we based it on participants of an ongoing cohort study (Group I), and obtained a very high response rate among those. Participation in the seroprevalence survey was therefore not biased by attitude to or experience with the pandemic. Group II individuals were also eligible to participate in the Rhineland Study. They had been invited in the past to take part in the Rhineland Study, but had not done so at the time. Now, during the pandemic, they actively approached us with the request whether they could still sign up and thereby then also participate in the SARS-CoV-2 seroprevalence study. Their willingness to become a participant in the Rhineland Study during the SARS-CoV-2 seroprevalence survey was thus likely biased by the pandemic and the incentive of being tested for the presence of SARS-CoV-2 antibodies. In the discussion section (lines 395-407) we use the difference between Group I and Group II seropositivity estimates to illustrate and warn against the substantial influence of self-referral / self-selection bias that we appropriately accounted for, but which generally is not considered in seroprevalence studies.

In order to better clarify this point in the manuscript, we have adapted the penultimate paragraph of the discussion section by including the following explanation:

"... Indeed, to the best of our knowledge, our serosurvey is the only available study in which the magnitude of self-selection bias has been specifically addressed with respect to SARS-CoV-2 seroprevalence. By presenting differences in the seropositivity rates for participants from Group I

(low risk of self-selection) and Group II (high risk of self-selection), who otherwise originated from the same pre-defined geographical region, our study can be used to illustrate and warn against the substantial influence of self-referral / self-selection bias in inflating seroprevalence estimates....“

Minor point 1:

1) Many abbreviations are not defined.

Response

We thank the reviewer for pointing this out. We have now closely checked the manuscript and have defined all unclear abbreviations.

Minor point 2:

2) Line 91: This sentence has a weird grammar structure.

Response

We have revised the sentence to read:

“The presence of antibodies that can neutralize the virus is highly specific for having sustained an infection and is also thought to constitute a major component of the acquired immunity to the virus.“

Minor point 3:

3) Line 376-380: Consider memory B cells, especially due to the long incubation time.

Response

We agree with the reviewer that a discussion of our findings in light of memory B (and T) cell immunity should be considered. Therefore, we have now complemented the discussion section with the following paragraph in this regard:

“...Although neutralizing antibodies are thought to be a major component of adaptive immunity and their levels also strongly correlate with the number of SARS-CoV-2 specific T-cells (Ni, Ye et al. 2020), a decline of neutralizing antibody levels does not necessarily imply absence of persisting immune memory (Cox and Brokstad 2020). Indeed, SARS-CoV-2 specific T-cells recognizing peptides derived from the spike, nucleoprotein and matrix components of the virus have also been found in convalescent plasma following mild COVID-19 and could confer a much longer lasting immunity (Grifoni, Weiskopf et al. 2020). Similarly, despite decreasing antibody levels after about three weeks post-symptom onset, a recent study found that the number of long-lived memory B cells in blood samples obtained from 25 convalescent COVID-19 patients continued to rise until about five months (Hartley, Edwards et al. 2020). Though there is still much to be learned about which components of the immune response provide protection against both initial infection and reinfection, recent evidence from SARS-CoV-2 vaccine trials support the notion that the presence and magnitude of neutralizing antibodies can be regarded as a valid marker of a protective immune response (Baden, El Sahly et al. 2020, Mulligan, Lyke et al. 2020, Walsh, Frenck et al. 2020). Therefore, the possibility cannot be excluded that even after infection a large proportion of individuals in the general population, especially those with asymptomatic or mild infections, may become susceptible again to SARS-CoV-2 infection.“

Reviewer #2

In this interesting study the authors undertake a community study and assess neutralising antibody status. The work is well performed with no technical concerns. The second cohort is somewhat less representative, smaller and likely to include many people who agreed to take part as they had symptoms. The statistical analysis is somewhat weakened by the small number of donors with neutralising antibodies.

This is a highly competitive area but the paper adds value in relation to presence of neutralising antibodies

- over time
- in relation to total antibody response
- symptoms

Response to Reviewer #2

We would like to thank the reviewer for her / his encouraging remarks and fully agree that, despite operating in a highly competitive field, our paper adds important new insights to this research area, especially with regard to study design and the correlates and temporal pattern of neutralizing antibodies to SARS-CoV-2.

As with regard to the comment regarding the second cohort, we would like to refer to our response to **Reviewer #1, major point 4**:

Group II did not bias our study because we analyzed Group II separately from Group I for calculating seroprevalence estimates. Rather, we included Group II to illustrate how seroprevalence studies that only start recruiting during the pandemic, are likely biased due to self-selection of participants.

The strength of our seroprevalence study is that we based it on participants of an ongoing cohort study (Group I), and obtained a very high response rate among those. Participation in the sero-prevalence survey was therefore not biased by attitude to or experience with the pandemic. Group II individuals were also eligible to participate in the Rhineland Study. They had been invited in the past to take part in the Rhineland Study, but had not done so at the time. Now, during the pandemic, they actively approached us with the request whether they could still sign up and thereby then also participate in the SARS-CoV-2 seroprevalence study. Their willingness to become a participant in the Rhineland Study during the SARS-CoV-2 seroprevalence survey was thus likely biased by the pandemic and the incentive of being tested for the presence of SARS-CoV-2 antibodies. In the discussion section (lines 395-407) we use the difference between Group I and Group II seropositivity estimates to illustrate and warn against the substantial influence of self-referral / self-selection bias that we appropriately accounted for, but which generally is not considered in seroprevalence studies.

In order to better clarify this point manuscript, we have adapted the penultimate paragraph of the discussion section by including the following elucidation:

"... Indeed, to the best of our knowledge, our serosurvey is the only available study in which the magnitude of self-selection bias has been specifically addressed with respect to SARS-CoV-2 seroprevalence. By presenting differences in the seropositivity rates for participants from Group I (low risk of self-selection) and Group II (high risk of self-selection), who otherwise originated from the same pre-defined geographical region, our study can be used to illustrate and warn against the substantial influence of self-referral / self-selection bias in inflating seroprevalence estimates...."

Reviewer #3

This is a well done study on an important area that is reported well. I just have a few comments about the presentation and interpretation of the results. My main comment refers to the interpretation of these results for herd immunity. I would like to see a greater discussion of T-cell immunity and the uncertainties about what is measured for antibody and T cell immunity and protection.

Response

We thank the reviewer for her / his favourable appraisal of our work and have incorporated all changes suggested, which we feel have substantially improved the presentation and interpretation of our findings. We have also included a more elaborate discussion of the uncertainties regarding the role of antibody and T-cell response in conferring immunity; please see our response to **Point 7** below.

Minor comments

Point 1:

Line 201: Is this number of symptom episodes or numbers of different symptoms described regardless of whether at same or different times? Please clarify.

Response

We have clarified this in the manuscript as follows:

"... In addition, we assessed the relation between the number of different symptoms experienced since January 1st, 2020 (regardless of whether at the same time or at different times), until the time of blood withdrawal and seropositivity..."

Point 2:

Line 225: I would suggest seroprevalence instead of prevalence as prevalence can be very easily confused with the prevalence of current infection.

Response

We agree and have replaced "Prevalence" with "Seroprevalence"

Point 3:

Line 321: Suggest removing "interestingly", not that it isn't interesting, but I don't think it is more interesting than the declines, given the study probably captured people at different times

after infection, an increase in some seems reasonable to me.

Response

We agree and have removed “interestingly” from this sentence.

Point 4:

Line 331: Please restate the time period for the seroprevalence estimates here.

Response

We have restated the time period for the seroprevalence estimates in this section.

Point 5:

Line 350: I’m not sure about the use of the word “evaded” this is to be expected given the asymptomatic nature of many infections. Perhaps this could be included in the discussion here.

Response

We have removed “evaded” and discuss our findings in this regard as follows:

“... Nevertheless, our findings suggest that a considerable number of individuals who had been infected went undetected by the local health regulatory agencies, most likely because of subclinical infection or development of mild symptoms only.” ...

Point 6:

Line 354: Suggest use of date rather than “current”.

Response

We have removed “current” from this sentence and make clear that we refer to the seroprevalence estimates as obtained in our serosurvey.

Point 7:

Line 376: I would suggest a greater discussion of T cell immunity, specifically here I think the correlation between antibody and T cell is still unclear. Generally I think the statement here and in the abstract about herd immunity is still too strong given these uncertainties.

Response

We agree with the reviewer that a more detailed discussion of our findings in light of memory T and B cell immunity would have been appropriate and our statements about herd immunity may have been too firm. Therefore, we have now:

1) Adapted the abstract, changing our initial statement in the last sentence to:

“Our findings indicate that in an unbiased sample, neutralizing antibodies are detectable in only one third of those with a positive immunoassay result, and wane relatively quickly, warranting further prospective in depth studies of population immunity.”

2) Added the following elaboration to the concerning section of the discussion, mindful of too strong statements:

“...Although neutralizing antibodies are thought to be a major component of adaptive immunity and their levels also strongly correlate with the number of SARS-CoV-2 specific T-cells (Ni, Ye et al. 2020), a decline of neutralizing antibody levels does not necessarily imply absence of persisting immune memory (Cox and Brokstad 2020). Indeed, SARS-CoV-2 specific T-cells recognizing peptides derived from the spike, nucleoprotein and matrix components of the virus have also been found in convalescent plasma following mild COVID-19 and could confer a much longer lasting immunity (Grifoni, Weiskopf et al. 2020). Similarly, despite decreasing antibody levels after about three weeks post-symptom onset, a recent study found that the number of long-lived memory B cells in blood samples obtained from 25 convalescent COVID-19 patients continued to rise until about five months (Hartley, Edwards et al. 2020). Though there is still much to be learned about which components of the immune response provide protection against both initial infection and reinfection, recent evidence from SARS-CoV-2 vaccine trials support the notion that the presence and magnitude of neutralizing antibodies can be regarded as a valid marker of a protective immune response (Baden, El Sahly et al. 2020, Mulligan, Lyke et al. 2020, Walsh, Frenck et al. 2020). Therefore, the possibility cannot be excluded that even after infection a large proportion of individuals in the general population, especially those with asymptomatic or mild infections, may become susceptible again to SARS-CoV-2 infection.”

REFERENCES

Baden, L. R., H. M. El Sahly, B. Essink, K. Kotloff, S. Frey, R. Novak, D. Diemert, S. A. Spector, N. Roupheal, C. B. Creech, J. McGettigan, S. Kehtan, N. Segall, J. Solis, A. Brosz, C. Fierro, H. Schwartz, K. Neuzil, L. Corey, P. Gilbert, H. Janes, D. Follmann, M. Marovich, J. Mascola, L. Polakowski, J. Ledgerwood, B. S. Graham, H. Bennett, R. Pajon, C. Knightly, B. Leav, W. Deng, H. Zhou, S. Han, M. Ivarsson, J. Miller and T. Zaks (2020). "Efficacy and Safety of the mRNA-1273 SARS-CoV-2 Vaccine." N Engl J Med.

Cox, R. J. and K. A. Brokstad (2020). "Not just antibodies: B cells and T cells mediate immunity to COVID-19." Nat Rev Immunol **20**(10): 581-582.

Grifoni, A., D. Weiskopf, S. I. Ramirez, J. Mateus, J. M. Dan, C. R. Moderbacher, S. A. Rawlings, A. Sutherland, L. Premkumar, R. S. Jadi, D. Marrama, A. M. de Silva, A. Frazier, A. F. Carlin, J. A. Greenbaum, B. Peters, F. Krammer, D. M. Smith, S. Crotty and A. Sette (2020). "Targets of T Cell Responses to SARS-CoV-2 Coronavirus in Humans with COVID-19 Disease and Unexposed Individuals." Cell **181**(7): 1489-1501.e1415.

Hartley, G. E., E. S. J. Edwards, P. M. Aui, N. Varese, S. Stojanovic, J. McMahon, A. Y. Peleg, I. Boo, H. E. Drummer, P. M. Hogarth, R. E. O'Hehir and M. C. van Zelm (2020). "Rapid generation of durable B cell memory to SARS-CoV-2 spike and nucleocapsid proteins in COVID-19 and convalescence." Sci Immunol **5**(54).

Huang, A. T., B. Garcia-Carreras, M. D. T. Hitchings, B. Yang, L. Katzelnick, S. M. Rattigan, B. Borgert, C. Moreno, B. D. Solomon, I. Rodriguez-Barraquer, J. Lessler, H. Salje, D. S. Burke, A. Wesolowski and D. A. T. Cummings (2020). "A systematic review of antibody mediated immunity to coronaviruses: antibody kinetics, correlates of protection, and association of antibody responses with severity of disease." medRxiv: Preprint. DOI: 2020.2004.2014.20065771.

Jahrsdörfer, B., J. Kroschel, C. Ludwig, V. M. Corman, T. Schwarz, S. Körper, M. Rojewski, R. Lotfi, C. Weinstock, C. Drosten, E. Seifried, T. Stamminger, H. J. Groß and H. Schrezenmeier (2020). "Independent side-by-side validation and comparison of four serological platforms for SARS-CoV-2 antibody testing." J Infect Dis.

Mulligan, M. J., K. E. Lyke, N. Kitchin, J. Absalon, A. Gurtman, S. Lockhart, K. Neuzil, V. Raabe, R. Bailey, K. A. Swanson, P. Li, K. Koury, W. Kalina, D. Cooper, C. Fontes-Garfias, P. Y. Shi, Ö. Türeci, K. R. Tompkins, E. E. Walsh, R. Frenck, A. R. Falsey, P. R. Dormitzer, W. C. Gruber, U. Şahin and K. U. Jansen (2020). "Phase I/II study of COVID-19 RNA vaccine BNT162b1 in adults." Nature **586**(7830): 589-593.

Ni, L., F. Ye, M. L. Cheng, Y. Feng, Y. Q. Deng, H. Zhao, P. Wei, J. Ge, M. Gou, X. Li, L. Sun, T. Cao, P. Wang, C. Zhou, R. Zhang, P. Liang, H. Guo, X. Wang, C. F. Qin, F. Chen and C. Dong (2020). "Detection of SARS-CoV-2-Specific Humoral and Cellular Immunity in COVID-19 Convalescent Individuals." Immunity **52**(6): 971-977.

Walsh, E. E., R. W. Frenck, Jr., A. R. Falsey, N. Kitchin, J. Absalon, A. Gurtman, S. Lockhart, K. Neuzil, M. J. Mulligan, R. Bailey, K. A. Swanson, P. Li, K. Koury, W. Kalina, D. Cooper, C. Fontes-Garfias, P. Y. Shi, Ö. Türeci, K. R. Tompkins, K. E. Lyke, V. Raabe, P. R. Dormitzer, K. U. Jansen, U. Şahin and W. C. Gruber (2020). "Safety and Immunogenicity of Two RNA-Based Covid-19 Vaccine Candidates." N Engl J Med **383**(25): 2439-2450.

Wölfel, R., V. M. Corman, W. Guggemos, M. Seilmaier, S. Zange, M. A. Müller, D. Niemeyer, T. C. Jones, P. Vollmar, C. Rothe, M. Hoelscher, T. Bleicker, S. Brünink, J. Schneider, R. Ehmann, K. Zwirgmaier, C. Drosten and C. Wendtner (2020). "Virological assessment of hospitalized patients with COVID-2019." Nature **581**(7809): 465-469.

REVIEWERS' COMMENTS

Reviewer #1 (Remarks to the Author):

The authors have addressed my questions well. However, the confusion for the PRNT50 and PRNT90 can easily be avoided by running dilutions, counting plaques and then doing an actual curve fit over the dilutions. The old-fashioned way this is done makes the results hard to understand for the vast majority of the community.

REVIEWERS' COMMENTS

Reviewer #1 (Remarks to the Author):

The authors have addressed my questions well. However, the confusion for the PRNT50 and PRNT90 can easily be avoided by running dilutions, counting plaques and then doing an actual curve fit over the dilutions. The old-fashioned way this is done makes the results hard to understand for the vast majority of the community.

Response to Reviewer #1:

We thank the reviewer for this comment. We intentionally performed a serial dilution of the participants' sera between 1:20 and 1:80 as the focus of the experiments was on assessing whether individuals lost their neutralizing antibodies during follow-up. Although we agree with the reviewer that running additional dilutions would allow for more precise estimation of PRNT50 values >1:80, this would not change any of our findings, because the cut-off threshold lies much lower at 1:20. Therefore, and also in light of the current situation and the enormous general work load at the Institute of Virology, we would respectfully chose not to pursue the suggestion for additional dilution experiments.